# Antitumor Effect of Regorafenib on MicroRNA Expression in Hepatocellular Carcinoma Cell Lines

**DOI:** 10.3390/ijms23031667

**Published:** 2022-01-31

**Authors:** Kei Takuma, Shintaro Fujihara, Koji Fujita, Hisakazu Iwama, Mai Nakahara, Kyoko Oura, Tomoko Tadokoro, Shima Mimura, Joji Tani, Tingting Shi, Asahiro Morishita, Hideki Kobara, Takashi Himoto, Tsutomu Masaki

**Affiliations:** 1Department of Gastroenterology and Neurology, Faculty of Medicine, Graduate School of Medicine, Kagawa University, 1750-1 Ikenobe, Miki-cho 761-0793, Kita-gun, Kagawa, Japan; takuma.kei@kagawa-u.ac.jp (K.T.); fujihara.shintaro@kagawa-u.ac.jp (S.F.); fujita.koji@kagawa-u.ac.jp (K.F.); nakahara.mai@kagawa-u.ac.jp (M.N.); oura.kyoko@kagawa-u.ac.jp (K.O.); tadokoro.tomoko@kagawa-u.ac.jp (T.T.); mimura.shima@kagawa-u.ac.jp (S.M.); tani.joji@kagawa-u.ac.jp (J.T.); shi.tingting@kagawa-u.ac.jp (T.S.); morishita.asahiro@kagawa-u.ac.jp (A.M.); kobara.hideki@kagawa-u.ac.jp (H.K.); himoto@chs.pref.kagawa.jp (T.H.); 2Life Science Research Center, Kagawa University, 1750-1 Ikenobe, Miki-cho 761-0793, Kita-gun, Kagawa, Japan; iwama.hisakazu@med.kagawa-u.ac.jp

**Keywords:** hepatocellular carcinoma, regorafenib, cell cycle, cell proliferation, cyclin, microRNA, antitumor effect

## Abstract

Hepatocellular carcinoma (HCC) is the most common primary malignancy of the liver and is one of the leading causes of cancer-related deaths worldwide. Regorafenib, a multi-kinase inhibitor, is used as a second-line treatment for advanced HCC. Here, we aimed to investigate the mechanism of the antitumor effect of regorafenib on HCC and evaluate altered microRNA (miRNA) expression. Cell proliferation was examined in six HCC cell lines (HuH-7, HepG2, HLF, PLC/PRF/5, Hep3B, and Li-7) using the Cell Counting Kit-8 assay. Xenografted mouse models were used to assess the effects of regorafenib in vivo. Cell cycle analysis, western blotting analysis, and miRNA expression analysis were performed to identify the antitumor inhibitory potential of regorafenib on HCC cells. Regorafenib suppressed proliferation in HuH-7 cell and induced G0/G1 cell cycle arrest and cyclin D1 downregulation in regorafenib-sensitive cells. During miRNA analysis, miRNA molecules associated with the antitumor effect of regorafenib were found. Regorafenib suppresses cell proliferation and tumor growth in HCC by decreasing cyclin D1 via alterations in intracellular and exosomal miRNAs in HCC.

## 1. Introduction

Hepatocellular carcinoma (HCC) is the most common primary malignant tumor of the liver [1] and is an important medical problem. In 2012, 782,000 cases were diagnosed worldwide, with 746,000 deaths, and the worldwide age-adjusted incidence rate was 10 cases per 100,000 person-years [2,3]. It is the second most common neoplasm and is ranked as the third leading cause of cancer-related deaths [2,3]. The incidence and mortality rate of HCC are decreasing in Asian countries; however, the number of new cases is continuously increasing in Western countries, and its incidence is increasing globally [4]. Standard systemic chemotherapy in patients with advanced HCC relies on the multi-kinase inhibitors sorafenib and lenvatinib, with regorafenib as the second-line therapy [5,6,7]. However, many patients do not respond to these treatments, and the overall survival rate of HCC remains poor [8]. Thus, novel and effective therapeutic strategies are of prime importance for advanced HCC.

Regorafenib is an oral multi-kinase inhibitor that targets vascular endothelial growth factor receptors (VGFR1, VEGFR2, and VEGFR3), platelet-derived growth factor receptors, fibloblast growth factor receptor 1 (FGFR 1), Raf, TIE-2, KIT, RET, and BRAF, which are protein kinases involved in angiogenesis, tumorigenesis, metastasis, and tumor immunity [9,10]. Regorafenib has been approved for the treatment of refractory metastatic colorectal cancer, advanced gastrointestinal stromal tumors, and refractory HCC [11,12]. Regorafenib induces cell cycle arrest and suppresses cell proliferation and tumor growth in various cancer cells, including lung squamous cell and bladder carcinomas, in in vitro and in vivo studies [13,14,15]. Regorafenib is expected to have an antitumor effect pathway, which is downstream of the main mechanism, cell cycle, and angiogenesis in HCC cells. However, the detailed mechanism of the antiproliferative effects of regorafenib in HCC cells via cell cycle and cell cycle-related molecules, including microRNA (miRNA), remains unclear.

The aim of this study was to confirm that regorafenib suppresses cell proliferation and tumor growth in HCC cells by inducing cell cycle arrest in vitro and in vivo, and to identify miRNA signatures associated with its antitumor effect. We examined the following: (1) the antitumor effects of regorafenib on HCC cell lines in vitro and in vivo; (2) the effects on the cell cycle and cell cycle-related molecules; and (3) miRNA signatures in HCC cells and exosomes.

## 2. Results

### 2.1. Regorafenib Suppresses Cell Proliferation of Human HCC Cells

We performed a cell proliferation assay using six human HCC cell lines (HuH-7, Hep3B, HepG2, Li-7, PLC/PRF/5, and HLF) to determine the antitumor effects of regorafenib in vitro. Regorafenib suppressed cell proliferation in a concentration-dependent manner in HuH-7, Hep3B, HepG2, Li-7 cells (Figure 1). Furthermore, it did not suppress growth in HLF, and PLC/PRF/5 cells at 3–5 µM that was the optimal concentration for experimental use, calculated from effective blood concentration of regorafenib in clinical cases. The HuH-7 cells were selected as to represent regorafenib-sensitive cells.

### 2.2. Regorafenib Induces Cell Cycle Arrest in the G0/G1 Phase and Affects Cell Cycle-Related Proteins

We evaluated the effect of regorafenib on the cell cycle of HuH-7 cells using flow cytometry (Figure 2A). For HuH-7 cells incubated with 5 μM regorafenib for 24–48 h, the number of cells in the S phase decreased and those in the G0/G1 phase increased compared to the untreated group (Figure 2B).

We further examined the effect of regorafenib by detecting G1/G0 arrest-associated proteins through western blotting. The expression of these proteins was compared between HuH-7 cells treated with 5 μM regorafenib at 24 or 48 h and controls. Consequently, we confirmed that cyclin D1, a key protein involved in the transition from the G0 phase to the G1 phase, showed decreased expression (Figure 2C). Regorafenib inhibited cyclin D1 expression in a dose-dependent manner. In addition, we analyzed other proteins involved in the G0 to G1 transition and found that CDK4 (the catalytic subunit of cyclin D1) levels decreased after the addition of regorafenib.

Together, these results indicate that regorafenib suppresses the G0/G1 phase transition and suppresses cell growth in HuH-7 cells by reducing cyclin D1.

### 2.3. Apoptosis-Inducing Effect of Regorafenib in HCC Cells

Apoptotic cells were detected using flow cytometry after regorafenib treatment. Figure 3 shows the proportion of live cells in the lower left quadrant, early apoptotic cells in the lower right quadrant, and late apoptotic HuH-7 cells in the upper right quadrant. The proportion of cells that underwent early apoptosis increased from 1.44% in the controls to 4.65% after 48 h of treatment with regorafenib in a representative experiment (Figure 3), but there was no significant difference between the proportion of cells that underwent early apoptosis in control and those treated with regorafenib in a three-repeat experiment.

### 2.4. Regorafenib Suppresses HCC Tumor Growth In Vivo

We conducted experiments to determine whether regorafenib has tumor-suppressive activity in HCC in vivo. We subcutaneously injected HuH-7 cells into nude mice and subsequently administered regorafenib orally. In the regorafenib treatment group, tumor growth was significantly suppressed by 46.6% on day 9 after administration compared to that in untreated mice (Figure 4). In addition, there was no significant difference in body weight between mice in the regorafenib-treated group and in the control group. No obvious effects other than tumor suppression were observed in these mice after regorafenib administration.

### 2.5. MiRNA Expression Signatures Are Different in Regorafenib-Treated and -Untreated HuH-7 Cells

We used a custom microarray platform to analyze the expression levels of 2555 miRNA probes. Unsupervised hierarchical clustering analysis showed that the regorafenib group was clustered separately from the control group (Figure 5A,B). Following normalization and removal of miRNAs with missing values, 133 significantly differentially expressed miRNAs were identified in regorafenib-treated HuH-7 cells, including 60 significantly upregulated miRNAs and 73 significantly downregulated miRNAs (Table 1). We also found 36 significantly differentially expressed miRNAs in the exosomes of regorafenib-treated cells, including 24 significantly upregulated miRNAs and 12 downregulated miRNAs (Table 2).

### 2.6. MiR-3714 Suppresses Cyclin D1 Expression in HCC Cells

Among the miRNAs differentially expressed on regorafenib treatment, four miRNAs (miR-494-3p, miR-3714, miR-4327, and miR-8073) were tumor suppressors associated with the decreased expression of cyclin D1 (Table 3).

The 4 miRNAs were quantified using RT-qPCR. miR-3714 was significantly upregulated in the regorafenib-treated group, compared to the untreated group, with a relative quantification (RQ) value of 6.85 ± 4.28 (*p* < 0.05) (Figure 6A). For miR-494-3p and miR-8073, the measured values varied from sample to sample, and no consistent trend could be found. miR-4327 did not yield stable results in the RT-qPCR and was not detectable in most of the samples.

To elucidate the role of miR-3714, which was identified as the most important miRNA by previous RT-PCR, we transfected HuH-7 cells with an miR-3714 mimic. Western blotting results confirmed that cells treated with the miR-3714 mimic exhibited a decrease in cyclin D1 expression, similar to that observed in cells treated with regorafenib (Figure 6B).

## 3. Discussion

The present findings are noteworthy as we investigated the antitumor effect of regorafenib on HCC growth, both in vitro and in vivo. Regorafenib induced cell cycle arrest at the G0/G1 phase by modulating the cell cycle-regulating protein cyclin D1 in the HuH-7 cells. These results are consistent with those of previous studies on the effects of regorafenib on various cancer cells [13,14,15]. Notably, the anti-proliferative effect of regorafenib on HCC was validated by modulating miRNAs in HCC cells and exosomes. To the best of our knowledge, this is the first study to show that regorafenib suppressed HCC cell proliferation by inducing cell cycle arrest and altered miRNA expression.

We examined the effect of regorafenib on cell proliferation in vitro using six different HCC cell lines. Our results reveal that regorafenib dose-dependently suppressed cell proliferation in HuH-7, Li-7, Hep3B, and HepG2 cells, but not in HLF, and PLC/PRF/5 cells. According to the Japanese collection of Research Bioresources Cell Bank (JCRB), HuH-7 cells are highly differentiated HCC producing Alpha-fetoprotein, PLC/PRF/5 cells are HBs antigen positive, and HLF cells are non-differentiated HCC. These difference in the characteristics of each cell line may have contributed to the differences in sensitivity to regorafenib, but in fact, there is limited evidence in the literature on the sensitivity and resistance of regorafenib to HCC. Since regorafenib has almost the same molecular structure as sorafenib with only fluorine bounded to it, the toxicity profile is similar, and previous studies reported that sorafenib resistance will help to elucidate regorafenib resistance. Expression of DNA methyltransferase (DMNT) in HCC is highly correlated with the expression of the octamer-binding transcription factor 4 (OCT4) and drug resistance. DMNT, OCT4, and IL-6 have been defined as predictive markers for HCC recurrence and poor prognosis [16]. In experiments using sorafenib-resistant HCC cells, DMNT has been suggested to have an important role in IL-6-mediated OCT4 expression and drug sensitivity in sorafenib-stimulated HCC. It has been reported that the activation level of STAT3 regulates DNMT/OCT4 and may be related to recurrence and poor prognosis of HCC patients [17].

Previous studies have reported that PLC/PRF/5 cells do not deactivate Akt signaling compared to Huh-7 cells [18]. It has also been reported that sorafenib exerts anticancer effects through inactivation of Akt signaling when premedicated on existing anticancer drugs [19]. Furthermore, HuH-7 cells, which highly expressed FGF19 and FGFR4, were more sensitive to lenvatinib, another type of tyrosine kinase receptor inhibitor (TKI) [20]. In contrast, the PLC/PRF/5 cells are resistant to pan-FGFR inhibitors [21,22]. Lenvatinib reported that FGF19 levels and lenvatinib susceptibility were correlated in HCC cell lines, and FGF19 inhibition eliminated lenvatinib susceptibility [23]. Thus, these endogenous mechanisms may be responsible for the difference in the antitumor effects of TKIs including regorafenib in Huh-7 and PLC/PRF/5.

Our in vitro experiments confirmed that regorafenib induced cell cycle arrest in the G0/G1 phase, which was accompanied by the downregulation of cyclin D1 in the HuH-7 cells. In PLC/PRF/5 cells that were not sensitive to regorafenib, the treatment of regorafenib did not alter the expression of cell cycle-related proteins, including cyclin D1 (Appendix A). As an activator of cell cycle progression, overexpression of cyclin D1 leads to dysregulation of CDK activity, the bypass of major cell checkpoints, unlimited cell proliferation, and tumor growth [24]. Several studies revealed that regorafenib has an antitumor effect, that is, it suppresses the cell cycle by inhibiting the expression of cyclin D1 [13,15]. Additionally, regorafenib has been shown to inhibit cell proliferation by inducing apoptosis in various cancer cell lines, such as lung squamous cell carcinoma [13] and bladder cancer [14]. However, in our present study, we could not confirm that regorafenib significantly induced apoptosis in HCC cells. The results indicate that regorafenib induced cell-cycle arrest and inhibited HuH-7 cell proliferation by suppressing cyclin D1 expression.

We analyzed miRNAs associated with the antitumor effects of regorafenib in HCC using miRNA expression arrays. miRNAs are small non-coding RNA molecules that can regulate gene expression by repressing translation or inducing mRNA degradation by hybridizing to the 3′-untranslated region of target mRNAs. It regulates the development and progression of cancer [25]. Previous studies have demonstrated that miRNA expression is associated with various cancers [25]. Several experimental evidences have also been shown to support the important role of miRNAs in HCC tumor growth [26,27,28].

According to the previous literature, there were a few reports showing the association of miRNAs with regorafenib treatment for HCC. Previously, we reported that serum miR-30d is a biomarker for predicting the efficacy of sorafenib therapy in hepatocellular carcinoma, indicating the optimal timing of switching to second-line therapy including regorafenib [29]. In an exploratory study, Teufel et al. analyzed plasma and tumor samples from study participants to identify miRNAs associated with response to regorafenib [30]. As a result, nine miRNAs (miR-30A, miR-122, miR-125B, miR-200A, miR-374B, miR-15B, miR-107, miR-320, miR-645) were identified in plasma, and their levels were significantly associated with overall survival to regorafenib. On the other hand, other authors have also reported that six down-regulated miRNAs (miR-101, -129-3p, -137, -149, -503 and -630) were correlate with worse clinical prognosis [31]. In addition, six miRNAs that promote metastasis of HCC (miR-29a, -219-5p, -331-3p, 425-5p, -487a and -1247-3p) are associated with poor clinical prognosis. Among these miRNAs, miR-1247-3p was also found to be downregulated by regorafenib in the exosome in our present study. Although there is no evidence from previous literature to suggest that miR-1247-3p is associated with cell cycle-related proteins including cyclin D1, the miRNA may be associated with the antitumor effect of regorafenib on HCC.

Our study also found that the expression of several miRNAs was significantly altered in HuH-7 cells after regorafenib treatment. Using microarray analysis, we identified 60 upregulated and 73 downregulated miRNAs in HuH-7 cells, as well as 24 upregulated and 12 downregulated miRNAs in exosomes in response to regorafenib treatment. It has been reported that some miRNAs, differentially expressed by regorafenib treatment, are tumor suppressors associated with decreased expression of cyclin D1 and have an antitumor effect on HCC. These include miR-494-3p, miR-3714, miR-4327, and miR-8073 (Table 3). MiR-8073 binds to at least five mRNAs (FOXM1, MBD3, CCND1, KLK10, and CASP2) in various cancers such as colon, pancreatic, and breast cancer in vitro and determines the gene and protein expression levels of these five targets. It has been reported to be negatively regulated to prevent tumor growth [27]. It has also been reported that administration of miR-8073 markedly suppresses the growth of colorectal tumors in xenograft mice in vivo [32]. miR-494-3p has been reported to be involved in the onset and progression of various carcinomas, such as lung and breast cancer [33,34]. miR-494-3p also contributes to tumor cell proliferation by activation of PI3K/AKT in HCC and is associated with poor prognosis in patients with HCC [35]. It has been reported that miR-3714 may be induced by the administration of quercetin (a major dietary flavonol) in endometriosis autoimplanted mouse models and is involved in the decreased expression of CCDN mRNA [36]. miR-3714 may also be involved in oxaliplatin resistance in colon cell carcinoma [37]. Although the roles played by some of these miRNAs are yet to be elucidated, these miRNAs may be associated with the antitumor effect of regorafenib in HCC. The mechanism of tumor suppression by miRNAs in HCC remains unclear, and the results of the findings of the current study have great impact.

The low sensitivity of HCC to chemotherapy, mostly due to multidrug resistance, and outcomes with TKIs including regorafenib, are not satisfactory for advanced HCC patients. In addition to therapeutic strategies with immune checkpoint inhibitors/antibody-based therapy and peptide-based vaccines, nucleic acid-based drugs such as miRNAs are promising therapeutic agents for HCC [38]. It was demonstrated that miR-4510 targeted several proto-oncogenes including GPC3 and RAF1, regulated key biological and signaling pathways including Wnt and RAS/RAF/MEK/ERK signals [39]. Specific miRNAs that act as tumor suppressors in the future could be applied to therapy using liposome-based miRNAs mimics.

This study has several limitations. First, HuH-7 cells were used as the main cell source of cells for some experiments in this study. Li-7 and Hep3B cells were not used in some experiments because the Li-7 cell line is mainly available in Japan but not in major cell banks around the world, and Hep3B cells are p53-deficient cells. Second, while we have presented the results of a microarray analysis of miRNAs in HuhH-7 cells, the functional analysis and study of the mechanism upstream of cyclinD1 are not sufficient and need to be further investigated in the future.

In conclusion, our study revealed that regorafenib suppresses HCC cell proliferation and HCC tumor growth, and it exerts antitumor effects by inducing cell cycle arrest in regorafenib-sensitive HCC cells.

## 4. Materials and Methods

### 4.1. Chemicals 

Regorafenib (ChemScene LLC, Monmouth Junction, NJ, USA) was purchased and prepared as a 10-mM stock solution in dimethyl sulfoxide (DMSO). 

### 4.2. Cell Lines and Culture 

Six human HCC cell lines, namely HuH-7, HepG2, PLC/PRF/5, HLF (the Japanese Cancer Research Bank, Osaka, Japan), Hep3B (the American Type Culture Collection, Manassas, VA, USA), and Li-7 (the Central Institute for Experimental Animals, Kanagawa, Japan), were used in this study. HuH-7 and PLC/PRF/5 cells were cultured in Dulbecco’s modified Eagle’s medium (DMEM) supplemented with 10% fetal bovine serum (FBS) and penicillin-streptomycin (100 µg/mL) in a humidified atmosphere containing 5% CO_2_. Similarly, we cultured HLF cells in DMEM supplemented with 5% FBS and penicillin-streptomycin. We also cultured HepG2 and Hep3B cells in MEM supplemented with 10% FBS, penicillin streptomycin, and Li-7 cells in Roswell Park Memorial Institute 1640 medium supplemented with 10% FBS and penicillin streptomycin.

### 4.3. Cell Proliferation Assay 

Cell proliferation assays were performed using the Cell Counting Kit-8 (CCK-8) (Dojindo Research Laboratories, Kumamoto, Japan) according to the manufacturer’s instructions. Each of the six cell lines was seeded in a 96-well plate (5.0 × 10^3^ cells/well) and cultured in 100 µL of medium corresponding to each cell line. After 24 h, cells were treated with regorafenib at 0, 1, 3, 5, or 10 µM concentrations and cultured for an additional 24–48 h. The medium was replaced with 100 µL of medium containing the CCK-8 reagent at each measurement time point, incubated for 3 h, and the absorbance was subsequently measured at 450 nm using an automatic microplate reader (Multiskan FC, Thermo Scientific Corp, Waltham, MA, USA).

### 4.4. Cell Cycle and Apoptosis Analysis 

To evaluate the growth inhibition mechanism, we performed flow cytometry analysis using the Cell Cycle Phase Detection Kit (Cayman Chemical Company, Ann Arbor, MI, USA). Regorafenib (5 µM) was added to HuH-7 cells (1.0 × 10^3^ cells/100 mm diameter dish) while DMSO was added as the control and incubated for 24–48 h. After washing with phosphate buffered saline (PBS), the HCC cells were stored at −20 °C until the following analysis. On the day of the flow cytometry experiments, cells were suspended in 100 µL PBS and 10 µL RNase A (250 µg/mL) and were incubated for 30 min.

Following regorafenib treatment, we analyzed apoptosis using flow cytometry and the Annexin V-FITC Early Apoptosis Detection Kit (Cell Signaling Technology, Boston, MA, USA). HuH-7 cells (1.0 × 10^6^ cells/100 mm-dimeter dish) were treated with DMSO as the control or 5µM regorafenib for 48 h. The proportion of early and late apoptosis cells were analyzed via double staining with FITC-conjugated Annexin V and propidium iodide (PI), according to the manufacturer’s instructions. The experiment was performed in triplicate, and the proportions of apoptotic cells were compared between the regorafenib-treated and untreated groups.

Flow cytometry was performed using a Cytomics FC 500 flow cytometer equipped with an argon laser (488 nm) (Beckman Coulter, Indianapolis, IN, USA). Cell percentages were analyzed using Kaluza software (ver.2.1) (Beckman Coulter).

### 4.5. Western Blot Analysis

Western blotting was performed in accordance with a previously described method [40]. HuH-7 cells (1.0 × 10^6^ cells/100 mm-dimeter dish) were treated with DMSO as the control or 5 μM regorafenib, and cultured for 24 and 48 h. Cells were lysed with a protease inhibitor cocktail: PRO-PREP Complete Protease Inhibitor Mix (iNtRON Biotechnology, Seongnam, Korea), the supernatant was stored at −80 °C until the following analysis. Briefly, proteins were subjected to sodium dodecyl sulfate polyacrylamide gel electrophoresis and subsequently transferred to nitrocellulose membranes. Following blocking, the membrane was incubated with the primary antibody to cyclin D1 (Thermo Fisher Scientific, Waltham, MA, USA), CDK4 and CDK2 (Santa Cruz Biotechnology, Dallas, TX, USA), β-actin (Sigma-Aldrich, St. Louis, MO, USA) and was further incubated with a horseradish peroxidase (HRP)-conjugated secondary antibody (Cell Signaling Technology, Danvers, MA, USA), which included HRP-conjugated anti-mouse and anti-rabbit IgG [41]. 

### 4.6. Analysis of the Therapeutic Effect of Regorafenib In Vivo 

Animal experiments (approval No. 19639) were conducted in accordance with the guidelines of the Kagawa University Laboratory Animal Committee and the ARRIVE guidelines set forth in the NC3Rs. Female athymic mice (BALB/c-nu/nu; 6 weeks old; 18–23 g) (Japan SLC, Shizuoka, Japan) were housed in a temperature-controlled environment under 12-h light and dark conditions. The HuH-7 cells (3.0 × 10^6^ cells per animal) were subcutaneously inoculated into the right flank of each mouse. Fourteen days later, 20 mice with masses > 3 mm in diameter were randomly assigned 1:1 to the control or treatment groups. We dissolved 0.6 mg regorafenib in saline containing 3% DMSO, and the solution was orally administered to mice (*n* = 10) in the treatment group five times a week [42,43]. Mice in the control group (*n* = 10) were administered the same volume of saline containing 3% DMSO orally. Mouse weight and tumor volume were recorded, and tumor volume (mm^3^) was calculated as (tumor length (mm) × tumor width (mm) × tumor width (mm))/2 [44]. One day after the start of treatment, one mouse from the treatment group was severely weakened and was sacrificed. Six days after the start of treatment, one mouse from the control group died in the cage. The remaining animals were sacrificed 9 days after starting treatment and data were compiled.

### 4.7. Microarray Analysis of MiRNAs

The HuH-7 cells were treated with 5 μM regorafenib or DMSO as the control for 48 h. Thereafter, total RNA was extracted using the miRNeasy Mini Kit (Qiagen, Venlo, The Nederland) according to the manufacturer’s instructions. Exosomal RNA was extracted using an exoRNeasy Serum/Plasma Maxi Kit (Qiagen). Following measurement of RNA quantity and quality using the RNA 6000 Nano Kit (Agilent Technologies, Santa Clara, CA, USA), samples were labeled using the miRCuRY Hy3 Power Labeling Kit (Exiqon A/S, Vedbaek, Denmark) and hybridized to a human miRNA Oligo Chip (v.21; Toray Industries, Inc., Tokyo, Japan). Scans were performed using a 3D-Gene Scanner 3000 (Toray Industries, Inc.). We used 3D-Gene extraction version 1.2 software (Toray Industries, Inc.) to calculate the raw signal strength of the images, and used GeneSpring GX 10.0 software (Agilent Technologies) to analyze raw data. Thereafter, differences in miRNA expression between regorafenib-treated and control samples were evaluated. Global level normalization was performed on the raw data above background levels. Differentially expressed miRNAs were determined using Welch’s *t*-test. We calculated the false discovery rate using the Benjamini-Hochberg method. Hierarchical clustering was performed using the farthest distance method, with the absolute decentered Pearson correlation coefficient as the metric. Heat maps were generated based on the relative expression intensity of each miRNA. The base-2 logarithm of the intensity was centered on the median of each row.

### 4.8. Quantitative Polymerase Chain Reaction (qPCR) Analysis of MiRNAs

For the 4 miRNAs (miR-494-3p, miR-3714, miR-4327, and miR-8073) that were identified as significant by the microarray analysis, we examined expression levels using real-time qPCR. Total RNA was extracted as described previously and diluted to 2.0 ng/µL. The TaqMan microRNA assay (Applied Biosystems, Waltham, MA, USA) was used to determine miRNA expression levels with U6 small nuclear RNA (RNU6B) as an internal control. miRNAs were reverse-transcribed using a TaqMan microRNA reverse transcription kit (Applied Biosystems). Reverse transcription was performed according to the manufacturer’s instructions. The reaction was denatured using a ViiA7 real-time PCR system (Applied Biosystems) via incubation at 95 °C for 20 s, followed by 40 cycles of 95 °C for 1 s, and 60 °C for 20 s. The relative miRNA expression levels were calculated using the comparative Ct method according to the following equation: 2 − ΔCt (ΔCt = miRCt − U6Ct).

### 4.9. Transfection of MiRNA Mimic

HuH-7 cells were transfected with the miR-3714 mimic (mirVana miRNA mimic, Ambion, Carlsbad, CA, USA) and Lipofectamine 2000 according to the manufacturer’s protocol.

### 4.10. Statistical Analyses

GraphPad Prism software version 8.0 (GraphPad Software, Inc., San Diego, CA, USA) was used for all statistical analyses. Data were analyzed using the unpaired *t*-test and *p* < 0.05 was considered statically significant.

## Figures and Tables

**Figure 1 ijms-23-01667-f001:**
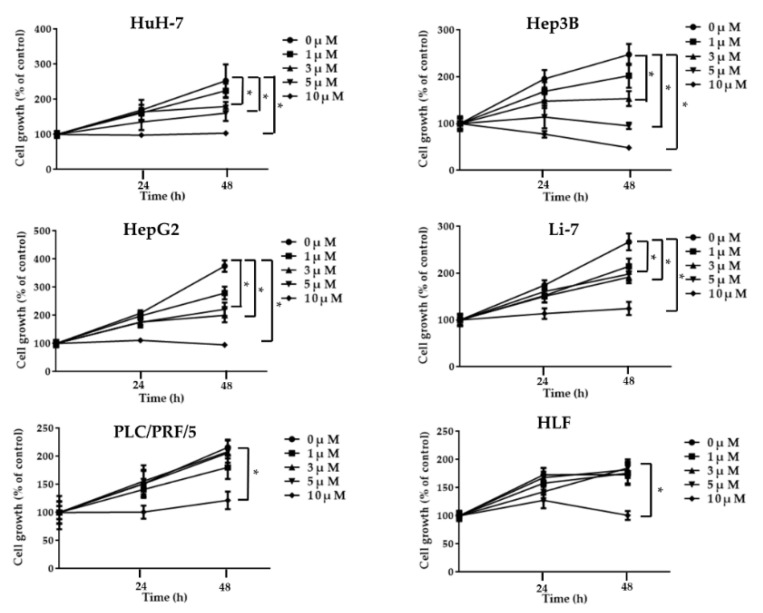
Cell proliferation assays. Regorafenib suppresses HCC cell proliferation. HuH-7, Hep3B, HepG2, Li-7, PLC/PRF/5, and HLF cells were treated with 0, 1, 3, 5, 10 µM regorafenib for 0, 24, and 48 h. Data points represent the mean cell number in three independent cultures, and error bars represent standard errors. For HuH-7, Hep3B, HepG2, Li-7 cell lines, concentration-dependent inhibition of cell proliferation was observed at 48 h after regorafenib treatment (* *p* < 0.01).

**Figure 2 ijms-23-01667-f002:**
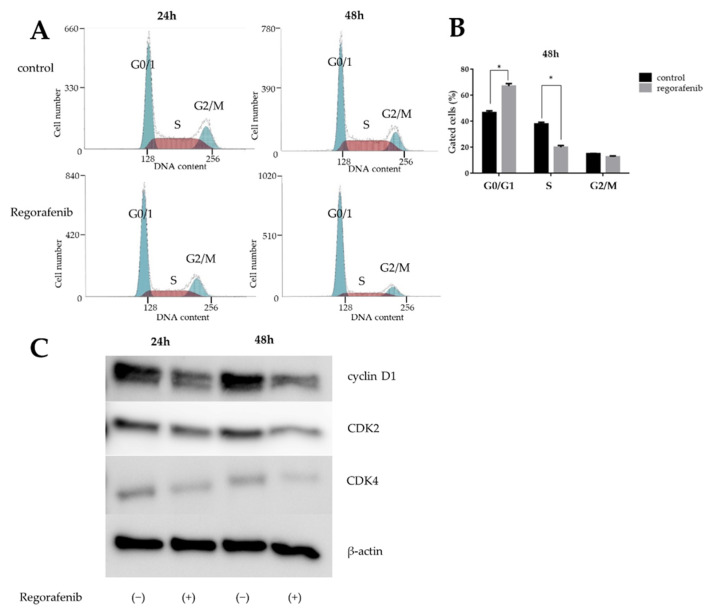
Regorafenib induces cell-cycle arrest at G0/G1 in HuH-7 cells. (**A**) Representative results showing the distribution of HuH-7 cells in G0/G1, S, and G2/M phases following treatment with 5 µM regorafenib at 24 and 48 h. (**B**) Histograms showing the proportion of HuH-7 cells in G0/G1, S, and G2/M phases (* *p* < 0.05). (**C**) Western blot showing the expression of cyclin D1, CDK2 and CDK4 in HuH-7 cells at 24 h and 48 h after the addition of 5 µM regorafenib.

**Figure 3 ijms-23-01667-f003:**
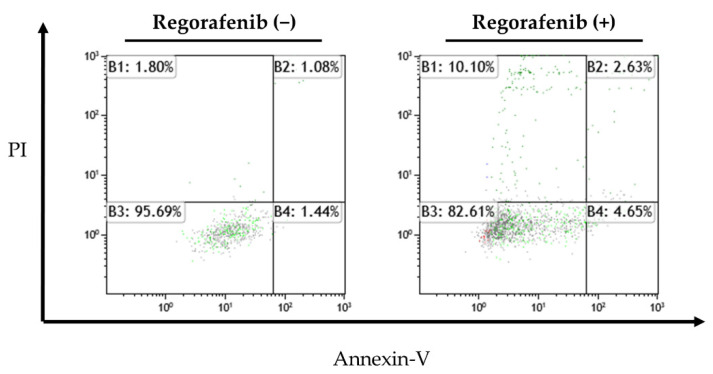
Apoptosis-inducing effect of regorafenib. Early apoptotic changes evoked by 5 µM regorafenib at 48 h were assessed by flow cytometry. Annexin-V positive and PI-negative cells were regarded as early apoptotic (enclosed areas in bold squares). The proportion of annexin V-positive and regorafenib-treated cells was slightly higher than that of untreated cells, but this difference was not significant.

**Figure 4 ijms-23-01667-f004:**
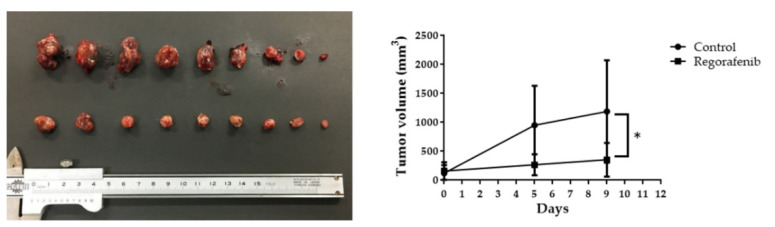
Antitumor effect of regorafenib on established HCC in vivo. The tumor volume (mm^3^) was calculated as follows: (tumor length (mm) × tumor width (mm) × tumor width (mm))/2. All animals were sacrificed on day 9 after treatment. Tumors were significantly smaller in regorafenib-treated mice than in control mice (* *p* < 0.05).

**Figure 5 ijms-23-01667-f005:**
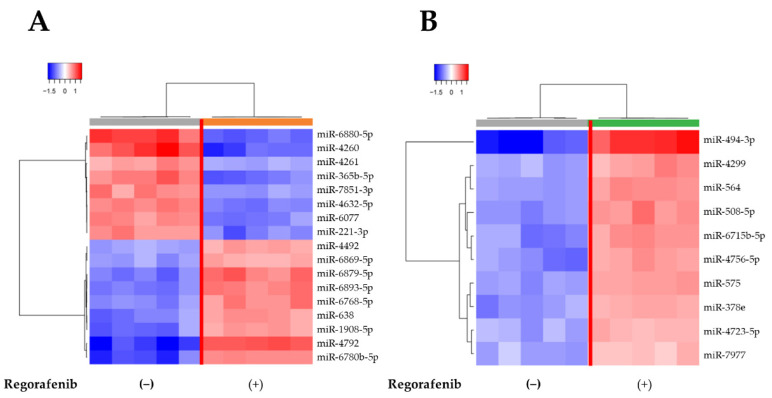
Hierarchical clustering of (**A**) HuH-7 cells and (**B**) HuH-7-derived exosomes cultured with or without 5µM regorafenib. The analyzed samples are presented in the columns, and the miRNAs are presented in the rows. The miRNA clustering color scale presented at the top of the figure indicates the relative miRNA expression levels, with red and blue representing high and low expression levels, respectively.

**Figure 6 ijms-23-01667-f006:**
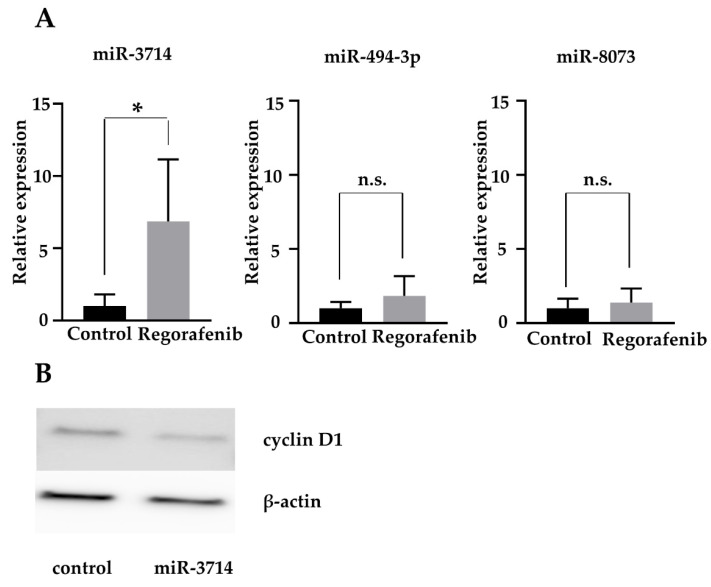
(**A**) Relative quantification (RQ) of miRNAs after treatment with or without 5 µM regorafenib for 48 h. miR-3714 expression was significantly upregulated by regorafenib treatment, while miR-494-3p and miR-8073 expression showed no significant changes (* *p* < 0.05) (n.s. not significant vs. control). (**B**) Western blot analysis of cyclin D1. The expression of cyclin D1 was attenuated at 48 h post-transfection with an miR-3714 mimic.

**Table 1 ijms-23-01667-t001:** Results of the statistical analysis of miRNAs in HuH-7 cells treated with or without regorafenib. Fold changes (FC) were determined as results in regorafenib-treated/untreated cells: FC > 2, FC < 0.5, *p* < 0.0001, false discovery ratio (FDR) < 0.05.

Upregulated				Down Regulated			
miRNA	FC	*p* Value	FDR	miRNA	FC	*p* Value	FDR
	(Treated/ Untreated)				(Treated/ Untreated)		
miR-6891-5p	13.915	0.000005	0.000080	miR-206	0.131	0.000025	0.000194
miR-3162-5p	12.868	0.000074	0.000400	miR-1292-5p	0.158	0.000007	0.000092
miR-3687	11.359	0.000001	0.000037	miR-4287	0.163	0.000035	0.000230
miR-3679-5p	9.705	0.000013	0.000126	miR-4633-5p	0.202	0.000002	0.000056
miR-4476	9.241	0.000014	0.000128	miR-4260	0.209	0.000000	0.000022
miR-4257	7.270	0.000001	0.000037	miR-4254	0.210	0.000004	0.000066
miR-6816-5p	7.039	0.000021	0.000171	miR-7843-5p	0.217	0.000015	0.000138
miR-6124	7.017	0.000007	0.000092	miR-6776-3p	0.219	0.000004	0.000065
miR-4725-3p	6.513	0.000012	0.000125	miR-6880-5p	0.223	0.000000	0.000003
miR-150-3p	6.180	0.000058	0.000339	miR-106b-3p	0.232	0.000016	0.000146
miR-3616-3p	6.046	0.000002	0.000044	miR-4319	0.240	0.000009	0.000109
miR-4294	5.425	0.000089	0.000462	miR-4695-3p	0.244	0.000004	0.000065
miR-4792	5.366	0.000000	0.000003	miR-4758-3p	0.247	0.000012	0.000125
miR-6870-5p	5.156	0.000002	0.000042	miR-874-5p	0.257	0.000027	0.000200
miR-885-3p	4.943	0.000051	0.000304	miR-6801-3p	0.261	0.000042	0.000264
miR-4749-5p	4.816	0.000001	0.000034	miR-887-3p	0.268	0.000034	0.000229
miR-365a-5p	4.812	0.000006	0.000082	miR-7114-5p	0.272	0.000026	0.000199
miR-1275	4.804	0.000004	0.000066	miR-4664-5p	0.283	0.000009	0.000109
miR-1229-5p	4.657	0.000089	0.000462	miR-5195-3p	0.287	0.000012	0.000125
miR-6721-5p	4.403	0.000003	0.000064	miR-365b-5p	0.298	0.000000	0.000011
miR-652-5p	4.156	0.000011	0.000125	miR-3177-3p	0.315	0.000002	0.000044
miR-1247-5p	4.036	0.000069	0.000377	miR-345-5p	0.316	0.000010	0.000111
miR-6511b-5p	4.019	0.000075	0.000400	miR-4687-3p	0.318	0.000005	0.000080
miR-6789-5p	3.907	0.000015	0.000138	miR-7155-5p	0.322	0.000033	0.000223
miR-6830-5p	3.768	0.000003	0.000061	miR-4632-5p	0.325	0.000000	0.000001
miR-6780b-5p	3.707	0.000000	0.000018	miR-6777-5p	0.325	0.000001	0.000034
miR-4327	3.460	0.000038	0.000247	miR-671-3p	0.328	0.000001	0.000034
miR-6879-5p	3.338	0.000000	0.000011	miR-1229-3p	0.331	0.000009	0.000111
miR-6893-5p	3.201	0.000000	0.000003	miR-5739	0.333	0.000003	0.000061
miR-1915-3p	3.136	0.000032	0.000223	miR-6735-5p	0.335	0.000005	0.000080
miR-3188	3.020	0.000039	0.000250	miR-6077	0.335	0.000000	0.000004
miR-1908-5p	2.890	0.000000	0.000018	miR-1913	0.339	0.000021	0.000171
miR-6768-5p	2.825	0.000000	0.000012	miR-221-3p	0.349	0.000000	0.000023
miR-4530	2.806	0.000003	0.000061	miR-187-5p	0.350	0.000001	0.000023
miR-3918	2.805	0.000020	0.000171	miR-6816-3p	0.363	0.000012	0.000125
miR-638	2.776	0.000000	0.000015	miR-4443	0.370	0.000004	0.000066
miR-4665-5p	2.769	0.000091	0.000465	miR-7851-3p	0.374	0.000000	0.000008
miR-663a	2.752	0.000010	0.000111	miR-6741-5p	0.378	0.000004	0.000066
miR-6798-5p	2.677	0.000031	0.000223	miR-4323	0.379	0.000031	0.000223
miR-6850-5p	2.652	0.000014	0.000128	miR-4786-3p	0.381	0.000067	0.000376
miR-6875-5p	2.651	0.000005	0.000069	miR-4786-5p	0.384	0.000098	0.000494
miR-4467	2.630	0.000003	0.000061	miR-320b	0.387	0.000017	0.000152
miR-6858-5p	2.472	0.000002	0.000044	miR-5002-3p	0.387	0.000009	0.000109
miR-4459	2.470	0.000004	0.000066	miR-128-1-5p	0.388	0.000076	0.000408
miR-6088	2.453	0.000034	0.000230	miR-3176	0.389	0.000001	0.000027
miR-149-3p	2.346	0.000039	0.000250	miR-4293	0.400	0.000035	0.000230
miR-4651	2.283	0.000013	0.000126	miR-6766-3p	0.401	0.000043	0.000265
miR-6869-5p	2.271	0.000000	0.000003	miR-320e	0.404	0.000024	0.000187
miR-4492	2.260	0.000000	0.000002	miR-885-5p	0.405	0.000030	0.000217
miR-3665	2.221	0.000013	0.000126	miR-6716-5p	0.410	0.000008	0.000103
miR-1202	2.189	0.000031	0.000223	miR-3135b	0.412	0.000022	0.000175
miR-6729-5p	2.151	0.000007	0.000089	miR-4502	0.423	0.000003	0.000064
miR-7111-5p	2.147	0.000013	0.000126	miR-3186-3p	0.425	0.000089	0.000462
miR-1237-5p	2.130	0.000004	0.000066	miR-4261	0.430	0.000000	0.000013
miR-5787	2.087	0.000002	0.000050	miR-675-5p	0.430	0.000018	0.000154
miR-6724-5p	2.058	0.000001	0.000037	miR-939-5p	0.437	0.000018	0.000154
miR-4756-5p	2.040	0.000069	0.000377	miR-320c	0.439	0.000030	0.000217
miR-6889-5p	2.032	0.000060	0.000342	miR-8052	0.446	0.000006	0.000082
miR-762	2.024	0.000064	0.000365	miR-1228-3p	0.448	0.000019	0.000162
miR-4508	2.017	0.000001	0.000025	miR-320a	0.450	0.000012	0.000125
				miR-320d	0.450	0.000046	0.000283
				miR-27b-3p	0.451	0.000089	0.000462
				miR-3154	0.458	0.000037	0.000239
				miR-6782-5p	0.463	0.000003	0.000061
				miR-4322	0.473	0.000026	0.000197
				miR-6859-3p	0.474	0.000091	0.000465
				miR-6515-3p	0.476	0.000013	0.000126
				miR-210-5p	0.477	0.000032	0.000223
				miR-4634	0.482	0.000027	0.000204
				miR-296-5p	0.487	0.000035	0.000230
				miR-4717-3p	0.491	0.000018	0.000156
				miR-4433a-5p	0.494	0.000012	0.000125
				miR-4498	0.495	0.000001	0.000037

**Table 2 ijms-23-01667-t002:** Results of statistical analysis of miRNAs in HuH-7-derived exosomes treated with or without regorafenib. Fold changes (FC) were determined as results in regorafenib-treated/untreated cells: FC > 2, FC < 0.5, *p* < 0.0001, false discovery ratio (FDR) < 0.05.

Upregulated				Down Regulated			
miRNA	FC	*p* Value	FDR	miRNA	FC	*p* Value	FDR
	(Treated/ Untreated)				(Treated/ Untreated)		
miR-494-3p	6.335	0.000000	0.000014	miR-4429	0.391	0.000014	0.000104
miR-933	5.926	0.000001	0.000023	miR-6511a-5p	0.408	0.000002	0.000031
miR-4502	4.810	0.000002	0.000030	miR-1247-3p	0.441	0.000012	0.000090
miR-2392	4.805	0.000016	0.000112	miR-5572	0.449	0.000085	0.000351
miR-1273c	4.457	0.000006	0.000063	miR-3180-3p	0.455	0.000011	0.000089
miR-6895-5p	3.346	0.000027	0.000156	miR-4448	0.473	0.000063	0.000276
miR-4468	3.332	0.000006	0.000063	miR-4322	0.479	0.000008	0.000073
miR-508-5p	2.788	0.000000	0.000003	miR-4649-5p	0.480	0.000005	0.000057
miR-6715b-5p	2.724	0.000000	0.000015	miR-4746-3p	0.494	0.000027	0.000156
miR-4756-5p	2.641	0.000000	0.000015	miR-3960	0.496	0.000036	0.000193
miR-564	2.583	0.000000	0.000000	miR-6820-5p	0.497	0.000007	0.000066
miR-491-5p	2.572	0.000006	0.000063	miR-6789-5p	0.499	0.000009	0.000078
miR-125a-3p	2.463	0.000009	0.000076				
miR-6893-5p	2.438	0.000009	0.000078				
miR-575	2.431	0.000000	0.000003				
miR-433-5p	2.395	0.000002	0.000036				
miR-378e	2.392	0.000000	0.000011				
miR-4299	2.379	0.000000	0.000015				
miR-6816-3p	2.350	0.000003	0.000040				
miR-4531	2.183	0.000004	0.000049				
miR-3714	2.094	0.000007	0.000066				
miR-4723-5p	2.072	0.000000	0.000014				
miR-3687	2.071	0.000008	0.000073				
miR-7977	2.014	0.000000	0.000015				

**Table 3 ijms-23-01667-t003:** Changes in miRNAs that were reported to suppress cyclin D1 expression on regorafenib treatment.

		Cell			Exosome		
miRNA	Chromosomal	FC	*p* Value	FDR	FC	*p* Value	FDR
	**Location**	(Treated/ Untreated)			(Treated/ Untreated)		
miR-494-3p	14q32.31	0.664	0.000687	0.002067	6.335	0.000000	0.000014
miR-3714	3p24.3	2.682	0.000613	0.001906	2.094	0.000007	0.000066
miR-4327	21q22.11	3.460	0.000038	0.000247	0.519	0.000001	0.000019
miR-8073	13q34	1.940	0.002027	0.004938	1.576	0.000033	0.000185

## Data Availability

All the data is presented in this article and in the Appendix A.

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
