# Peer review of "Antitumor Effect of Regorafenib on MicroRNA Expression in Hepatocellular Carcinoma Cell Lines"

_ijms, 2022, doi:10.3390/ijms23031667_

Round 1

Reviewer 1 Report

Manuscript "Antitumor effect of regorafenib on microRNA expression in hepatocellular carcinoma cell lines" is an interesting scientific study with a clinical aspect. The research concerns the current issue of difficult-to-treat hepatocellular carcinoma.
This paper is properly drafted. Each chapter is synthetically and exhaustively prepared. It is very important to use two experimental models for research, ie 6 human HCC cell lines and thirty female athymic mice. The test results are presented clearly and in detail. In the References chapter, the Authors included relevant and up-to-date publications, the mostly published in the last 10 years.
Due to the high scientific and clinical significance, I propose to publish the paper "Antitumor effect of regorafenib on microRNA expression in hepatocellular carcinoma cell lines" in the International Journal of Molecular Sciences in the present form.

Reviewer 2 Report

The manuscript by Takuma et al. aims at demonstrating the antitumor effect of regorafenib on microRNA expression in hepatocellular carcinoma cell lines. Overall, the article looks promising; however some major points need to be addressed to improve its quality:

1) The effects of regorafenib should be tested on at least one more cell line in order to validate what has been observed in HuH-7 cells.

2) What is the effect of regorafenib on cyclin D3, p21 and p27? And on caspase 9, 8 and 3 and PARP? How does the expression of these proteins change with respect to miR-3714?

3) What is the effect of miR-3714 silencing on regorafenib efficacy in hepatocellular carcinoma cells?

4) Why did the authors select miR-3714 for their experiments? This should be better discussed.

Round 2

Reviewer 2 Report

Despite being promising, the manuscript by Takuma et al. has too many weak points. The use of only one cell line and the absence of a correlation between miR-3714 expression and cell proliferation makes the conclusions reached by the authors speculative. Therefore, I am sorry to recommend this article for rejection.

Author Response

Thank you for your comments. We have revised the manuscript further based on the feedback from the academic editor. Please refer to the attached word file.

Round 3

Reviewer 2 Report

The manuscript has been improved and can be accepted for publication in IJMS.